# BICD Cargo Adaptor 1 (BICD1) Downregulation Correlates with a Decreased Level of PD-L1 and Predicts a Favorable Prognosis in Patients with IDH1-Mutant Lower-Grade Gliomas

**DOI:** 10.3390/biology10080701

**Published:** 2021-07-22

**Authors:** Shang-Pen Huang, Chien-Hsiu Li, Wei-Min Chang, Yuan-Feng Lin

**Affiliations:** 1Center of General Education, Chung Hua University, Hsinchu 707, Taiwan; neuro1471147@yahoo.com.tw; 2Department of Neurology, Po-Jen General Hospital, Taipei 105, Taiwan; 3Genomics Research Center, Academia Sinica, Taipei 11529, Taiwan; yungchieh.c@gmail.com; 4Graduate Institute of Clinical Medicine, College of Medicine, Taipei Medical University, Taipei 11031, Taiwan; 5Department of Law, School of Law, Ming Chuan University, Taipei 111, Taiwan; 6School of Oral Hygiene, College of Oral Medicine, Taipei Medical University, Taipei 11031, Taiwan; 7Cell Physiology and Molecular Image Research Center, Wan Fang Hospital, Taipei Medical University, Taipei 11696, Taiwan

**Keywords:** *BICD1*, lower-grade gliomas, biomarker, prognosis, *PD-L1*

## Abstract

**Simple Summary:**

The hypoxic inducible factor 1A (HIF1A) pathway has been known to play an important role in tumor progression in various cancers, including lower-grade (Grade II/III) gliomas (LGGs). An *in silico* analysis using 34 genes associated with the activity of the HIF1A pathway demonstrated that the BICD cargo adaptor 1 (BICD1) gene is a potential prognostic marker in LGGs. Moreover, BICD1 gene (*BICD1*) expression was positively correlated with *CD274*, *GSK3B*, *HGF*, and *STAT3* expression in LGGs. Importantly, *BICD1* downregulation was significantly associated with well-known favorable prognostic markers, such as a higher Karnofsky performance score (KPS), *IDH1/TP53/ATRX* mutations, wild-type *EGFR* and younger patient age, in LGGs. Therefore, our findings present BICD1 as a new prognostic biomarker to more precisely predict the clinical outcomes of LGG patients in coordination with those well-known biomarkers.

**Abstract:**

Although several biomarkers have been identified to predict the prognosis of lower-grade (Grade II/III) gliomas (LGGs), we still need to identify new markers to facilitate those well-known markers to obtain more accurate prognosis prediction in LGGs. Bioinformatics data from The Cancer Genome Atlas (TCGA), the Chinese Glioma Genome Atlas (CGGA), and the Cancer Cell Line Encyclopedia (CCLE) datasets were used as the research materials. In total, 34 genes associated with the HIF1A pathway were analyzed using the hierarchical method to search for the most compatible gene. The BICD cargo adaptor 1 (BICD1) gene (*BICD1*) was shown to be significantly correlated with The hypoxic inducible factor 1A (HIF1A) expression, the World Health Organization (WHO) grade, and *IDH1* mutation status. In addition, *BICD1* downregulation was significantly correlated with a higher Karnofsky performance score (KPS), *IDH1*/*TP53*/*ATRX* mutations, wild-type *EGFR,* and younger patient age in the enrolled LGG cohort. Moreover, *BICD1* expression was significantly upregulated in wild-type *IDH1* LGGs with *EGFR* mutations. Kaplan–Meier survival analysis revealed that *BICD1* downregulation predicts a favorable overall survival (OS) in LGG patients, especially in those with *IDH1* mutations. Intriguingly, we found a significant correlation between *BICD1* downregulation and a decreased level of *CD274*, *GSK3B*, *HGF*, or *STAT3* in LGGs. Our findings suggest that BICD1 downregulation could be a potential biomarker for a favorable prognosis of LGGs.

## 1. Introduction

Lower-grade gliomas (LGGs), defined by the World Health Organization (WHO) as grades II and III gliomas, include oligodendrogliomas and astrocytomas [1]. There is a wide range of the 5-year survival rate in LGG patients because the clinical behaviors of LGGs are highly variable [2]. Stratification of LGGs into more clinically distinct subgroups may help clinicians more accurately predict patients’ prognosis. In the past, the WHO grade, histological type of tumor, and patient age were used to classify LGG patients and predict their prognosis [3]. However, the WHO histopathological classification still remains unsatisfactory and limited because of the inter-observer variability and low objectivity in reading the histopathological data of LGG samples [4].

To overcome this limitation, several well-known molecular markers, including chromosome 1p19q codeletions, and mutations in *IDH1, TP53*, *ATRX*, *CIC*, and *FUBP1*, and the promoter of *TERT* have been reported for diagnosis and classification of LGGs [5,6,7]. Therefore, the 2016 WHO classification of tumors of the CNS system incorporated those molecular parameters with classical histopathological subtypes of LGGs to make a new guideline for the classification of LGGs [8]. Although the *IDH1* mutation status and 1p19q codeleted status are the most well-known biomarkers of LGGs and have been validated as powerful and useful prognosis indicators for LGG patients, there is still a wide range of clinical outcomes even in patients who have the same *IDH1* or 1p19q codeleted status [9,10]. Hence, we aimed to search for a new molecular marker that can facilitate *IDH1* status to classify LGGs into further clinically distinct subgroups, which may provide more accurate prognosis prediction for LGG patients.

Hypoxia has already been shown to correlate with poor prognosis and can be used to predict survival and response to therapy in multiple cancer models [11]. Several studies have demonstrated that hypoxia inducible factor 1A (HIF1A) and its downstream genes (e.g., *GLUT1*, *VEGF*, *CA9*) are also poor prognostic markers for various cancers [12]. In order to detect hypoxic conditions in cancer tissue, most previous studies focused on targeting *HIF1A*. However, measurement of the HIF1A protein is somehow difficult due to its instability under normoxic conditions. Immunohistochemistry (IHC) is currently used to measure the protein level of HIF1A in a hypoxic cancer tissue, but it needs invasive biopsy of the patient’s tumor and lacks reproducibility and consistency due to inter-observer variability. In addition, biopsy samples obtained from different sites of the primary tumor would might have different hypoxic conditions. All of these would limit the clinical application of targeting the HIF1A protein for predicting the prognosis of cancer patients.

Gene sequencing could be an affordable and reliable approach to get prognostic information from patients’ tumor samples. However, the gene expression of HIF1A did not show prognostic significance in the Cancer Genome Atlas (TCGA) LGG cohort (Appendix A). Therefore, we searched for genes associated with the HIF1A pathway as research materials for this study. We selected 34 genes, which were reported to be associated with the HIF1A pathway [13,14,15], as candidate genes for further analyses (Appendix A). These genes will be screened according to the criteria we proposed for this study: significantly and positively correlated with HIF1A expression (upregulation), the WHO grade (grade III), and *IDH1* mutation status (wild-type). The gene most compatible with our criteria will be selected as the putative marker for further correlation and survival analyses, and its prognostic significance and correlations with the clinicopathological features and important markers of LGGs will be validated in LGG patients in the Chinese Glioma Genome Atlas (CGGA) dataset and in LGG cell lines in the Cancer Cell Line Encyclopedia (CCLE) dataset.

## 2. Materials and Methods

### 2.1. Clinicopathological Data and Gene Expression Profiles of Patients with LGGs from the TCGA Database

The clinicopathological data of patients in the TCGA LGG cohort were downloaded from the TCGA Portal (http://xena.ucsc.edu/welcome-to-ucsc-xena/, accessed on 1 February 2021). Patients’ clinicopathological information, including gender, age at initial diagnosis, the Karnofsky performance score, the WHO grade, histological subtypes, the overall survival time, and survival status, were collected from the aforementioned website.

Gene sequencing data, including the gene expression levels (detected by gene expression RNAseq (IlluminaHiSeq, San Diego, CA, USA), mutation status of specific genes (detected by somatic mutation (SNP and INDEL)), and copy number variation in specific chromosomes (detected by copy number segments), were also downloaded from the above TCGA Portal.

### 2.2. Clinicopathological Data and Gene Expression Profiles of Patients with LGGs from the CGGA Database

LGG patients in the CGGA dataset (*n* = 420) were used as a validation cohort. The clinicopathological information and gene sequencing data, including the WHO histological grade, the overall survival time, survival status, *IDH1* mutation status, chromosome 1p19q codeleted status, and the expression levels of specific genes, were downloaded from the CGGA website (http://www.cgga.org.cn/).

### 2.3. Gene Sequencing Data of LGG Cell Lines in the CCLE Dataset from the TCGA Database

The TCGA website (http://xena.ucsc.edu/welcome-to-ucsc-xena/, accessed on 1 February 2021) also provided the gene sequencing data of glioma cell lines, including LGGs and glioblastomas, in the CCLE dataset. All the cell lines we used in this manuscript were rechecked in the website of Cancer Model Passport (https://cellmodelpassports.sanger.ac.uk/) to ensure all of them are LGG cell lines. The expression levels of BICD1 and *CD274* (PD-L1) (detected by gene expression RNAseq) in the eight LGG cell lines in the CCLE dataset were downloaded from the above TCGA Portal.

### 2.4. Subgroups of Patients with LGGs for Further Analyses

LGG patients in the TCGA LGG cohort (*n* = 508) and in the CGGA dataset (*n* = 420) were stratified into subgroups for correlation and survival analyses.

For correlation analyses, patients were stratified into two subgroups according to their clinicopathological features, including age (>40 vs. ≤40), gender (male vs. female), the WHO grade (grade III vs. grade II), histological subtypes (astrocytoma vs. others), the mutation status (mutant vs. wild-type) of specific genes (including *IDH1, TP53, ATRX* and *EGFR*), chromosome 1p19q codeleted status (codeleted vs. others), the Karnofsky performance score (≤80 vs. >80), and the expression levels of BICD1 (50% high vs. 50% low). For the age factor, the age of 40 was determined as the cutoff value, by which patients were stratified into two approximately equal groups, the younger (age≤40, *n* = 249, 49%) and the older (age>40, *n* = 259, 51%).

For survival analyses, patients were stratified into two subgroups by their clinicopathological features, the mutation status of specific genes (including *IDH1, EGFR*) and the expression levels of BICD1 (50% high vs. 50% low, 33.1% high vs. 66.9% low, 20.1% high vs. 79.9% low, and 10.0% high vs. 90.0% low, respectively).

### 2.5. Statistical Analysis

All statistical analyses were performed with the use of SPSS version 20.0 software (SPSS, Chicago, IL, US). The scatter pictures and dot plots were drawn with the use of Prism 5 software (GraphPad software Inc., San Diego, CA, US). A *t* test was used to analyze the differences in gene expression levels between different subgroups of LGG patients. Correlations between downregulation of BICD1 and the clinicopathological features of LGG patients were analyzed by Pearson’s Chi-square test. The overall survival time and 5-year survival rate of patients in different LGG subgroups were calculated using the Kaplan–Meier method, and the difference in the overall survival between subgroups of patients were compared by the log-rank test. The survival curves of patients in different LGG subgroups were drawn with the use of SPSS software. To identify the factor which had a significant and independent impact on patients’ overall survival, BICD1 downregulation and the clinicopathological features of LGGs, which were potentially related to patients’ survival, were compared by univariate and multivariate analyses with the Cox proportional hazard model. For all analyses, a *p* value of <0.05 was considered statistically significant.

## 3. Results

### 3.1. Identification of 12 Genes from HIF1A-Associated Pathways as Candidate Markers of LGGs

In total, 34 genes reported to be associated with the hypoxia inducible factor 1A (HIF1A) pathway (Appendix A) were analyzed and ranked according to the criteria we proposed for this study: significantly and positively correlated with HIF1A expression (upregulation), the WHO grade (grade III), and *IDH1* mutation status (wild-type). The expression heatmaps of genes associated with the HIF1A pathway were constructed according to the gene sequencing data obtained from the TCGA LGG cohort (*n* = 508). There were 18 genes whose expression levels were significantly and positively correlated with HIF1A upregulation (Figure 1A). There were 26 genes whose expression levels were significantly and positively correlated with a higher WHO grade (grade III) (Figure 1B), and there were 28 genes whose expression levels were significantly and positively correlated with *IDH1* status (wild-type) (Figure 1C). The hierarchical method revealed a total of 12 genes, i.e., *FN1*, *CD274* (programmed death-ligand 1, PD-L1), *CXCR4*, *BICD1*, *IGFBP3*, *DDIT4*, *VEGFA*, *PDCD1* (programmed cell death-1, PD-1), *TGFB1*, *TGFB3*, *VIM*, and *HGF*, compatible with our criteria (Figure 1D). The 12 genes were selected as candidate markers for further analyses.

### 3.2. BICD1 Gene Severs as a Potential Prognostic Marker of LGGs 

Based on the heatmap analysis, the 12 candidate markers compatible with our criteria were compared with the clinicopathological features and several important molecular markers of LGGs for evaluating their prognostic significance in LGGs (Figure 2A). In patients with dead status in the TCGA LGG cohort, there were apparent correlations of a shorter survival time (whose survival time is less than the median value of the survival time in all LGG patients) with an older patient age, a higher WHO grade (grade III), the astrocytoma subtype of LGGs, wild-type *IDH1/TP53/ATRX*, mutations in *EGFR*, upregulation of *BICD1*, *VIM*, *IGFBP3*, *PDCD1* (PD-1), *HGF*, *FN1*, *VEGFA*, *CD274* (PD-L1), *TGFB3*, *CXCR4* and *TGFB1*, downregulation of *DDIT4*, and chromosome 1p19q codeletions.

Based on the Cox regression analysis, the 12 candidate markers, as well as the clinicopathological features of patients in the TCGA LGG cohort, were ranked according to their hazard ratio (HR) (Figure 2B). Undoubtedly, those well-known poor prognostic markers of LGGs, including *EGFR* status (mutant vs. wild-type, HR = 5.162), *IDH1* status (wild-type vs. mutant, HR = 4.445), the WHO grade (grade III vs. grade II, HR = 3.314) and patient age (>40 vs. ≤40, HR = 2.889), showed the highest HRs. The expression of BICD1 gene (*BICD1*, High vs. Low) also demonstrated the highest HR (2.731) among the 12 candidate genes. 1p19q codeleted status (Others vs. Codeleted) also presented a high HR (2.602). However, its HR was not as high as that of *BICD1* expression and other well-known markers of LGGs (Figure 2B). Therefore, the BICD1 gene was chosen as the putative marker of LGGs for further correlation and survival analyses in this study.

### 3.3. BICD1 Downregulation Correlates with Benign Clinicopathological Features of LGG Patients

In this study, we demonstrated that downregulation of *BICD1* was significantly correlated with a couple of benign clinicopathological features of patients in the TCGA LGG cohort (Appendix A). The heatmap revealed that downregulation of the BICD1 gene was more enriched in grade II gliomas than in grade III gliomas, and the *t*-test analysis revealed that the expression levels of *BICD1* were significantly lower in grade II gliomas than in grade III gliomas. However, in the histological subtypes of LGGs, including oligodendroglioma and astrocytoma, *BICD1* was not differentially expressed (Figure 3A).

The Karnofsky Performance Score (KPS) is used for evaluating a patient’s clinical performance. In the TCGA LGG cohort (*n* = 300), the heatmap revealed that downregulation of *BICD1* was enriched in patients with their KPS ≥ 90, and in the *t*-test analysis, the expression levels of *BICD1* were significantly lower in patients with KPS = 100 than in those with KPS = 80 (*p* = 0.0454) or KPS ≤ 60 (*p* = 0.0029) (Figure 3B). The correlations between the KPS and the clinicopathological features of patients in the TCGA LGG cohort (*n* = 300) were analyzed. *IDH1* mutations, wild-type *EGFR*, and downregulation of *BICD1* showed significant correlations with a higher KPS (≥90) of LGG patients (Appendix A).

*IDH1* mutation status and chromosome 1p19q codeleted status are important markers for the classification of LGGs. In the TCGA LGG cohort (*n* = 508), the heatmap revealed that upregulation of *BICD1* was enriched in wild-type *IDH1* LGGs without 1p19q codeletions, which also had profound *EGFR* mutations and dead status of patients. Interestingly, mutations in *TP53* and *ATRX* were enriched in LGGs without 1p19q codeletions, especially in those with *IDH1* mutations (Figure 4A). In *t*-test analyses, the expression levels of *BICD1* were significantly lower in LGGs with mutant *IDH1*, *TP53*, or *ATRX*. Its expression levels were also significantly higher in LGGs with *EGFR* mutations (Figure 4B).

The expression levels of *BICD1* were not significantly correlated with 1p19q codeleted status in LGGs. When LGGs were stratified into four subgroups depending on their *IDH1* status, 1p19q codeleted status and *EGFR* status, *BICD1* was differentially expressed within the four subgroups, and its expression levels were highest in the subgroup of LGGs with wild-type *IDH1* and mutant *EGFR*, which could be considered to be glioblastoma (grade IV glioma), rather than LGGs (grade II and III gliomas) according to the 2016 WHO classification of CNS tumors (Figure 4C). In addition, the expression levels of *BICD1* were significantly and positively correlated with patient age and were significantly lower in LGG patients with alive status. However, its expression levels were not significantly distinct between males and females (Figure 4D).

### 3.4. BICD1 Downregulation in Coordination with IDH1 or EGFR Status Effectively Predicts a Faborable Overall Survival of LGG Patients

To evaluate the prognostic significance of *BICD1*, patients in the TCGA LGG cohort (*n* = 508) were stratified into two groups according to their *BICD1* expression for Kaplan–Meier survival analyses (Figure 5A). When LGG patients were stratified by their *BICD1* expression (50% high, *n* = 254 vs. 50% low, *n* = 254), those with upregulation of *BICD1* had a significantly worse overall survival (OS) (median survival: 4.827 years, 5-year survival: 49.1%) (*p* = 5.437 × 10^−7^). When patients were stratified by their *BICD1* expression (33.1% high, *n* = 168 vs. 66.9% low, *n* = 340), those with upregulation of *BICD1* had a more significantly worse OS (median survival: 3.427 years, 5-year survival: 39.8%) (*p* = 2.941 × 10^−9^). When patients were stratified by their *BICD1* expression (20.1% high, *n* = 102 vs. 79.9% low, *n* = 406), those with upregulation of *BICD1* had a further significantly worse OS (median survival: 2.830 years, 5-year survival: 32.2%) (*p* = 2.904 × 10^−12^). When patients were stratified by their *BICD1* expression (10% high, *n* = 51 vs. 90% low, *n* = 457), the difference in the OS between the two groups of LGGs had the highest significance (*p* = 3.532 × 10^−14^), and those with upregulation of *BICD1* had the worst OS (median survival: 1.868 years, 5-year survival: 20.6%) (Figure 5A). Our analyses confirmed that upregulation of *BICD1* had a significant impact on the worsening of OS of LGG patients, and downregulation of *BICD1* had a significant impact on a favorable outcome of LGG patients, and those with the highest expression of *BICD1* had the worst prognosis (the shortest survival time and the lowest 5-year survival rate).

Consistent with previous studies that the *IDH1* mutation is a favorable prognostic marker for LGGs, LGG patients with *IDH1* mutations (*n* = 394) had a significantly better OS (median survival: 7.964 years, 5-year survival: 71.1%) than those with wild-type *IDH1* (*n* = 114) (median survival: 2.077 years, 5-year survival: 29.2%) (Figure 5B), which had the highest statistical significance (*p* = 2.823 × 10^−18^) compared with other well-known markers (Figure 2B). Conversely, LGG patients with *EGFR* mutations (*n* = 35) had a worse OS (median survival: 1.532 years, 5-year survival: 7.1%) than those with wild-type *EGFR* (*n* = 473) (median survival: 7.877 years, 5-year survival: 67.0%), which also had a high statistical significance (*p* = 2.116 × 10^−14^) (Figure 5C). *EGFR* status (mutant vs. wild-type) also had the highest impact (HR = 5.162) on a worse OS of LGG patients (Figure 2B).

To improve the prognostic accuracy of the *IDH1* and *EGFR* mutation status, we incorporated *BICD1* expression with these well-known markers to stratify LGG patients into more distinct subgroups for further survival analysis. Patients in the TCGA LGG cohort (*n* = 508) were stratified into four subgroups according to their *IDH1* status and *BICD1* expression (Figure 5D). In LGG patients with *IDH1* mutations (*n* = 394, 77.6%), those with downregulation of *BICD1* (*n* = 230, 45.3%) (median survival: 8.186 years, 5-year survival: 80.3%, adjusted HR = 0.452) had a significantly better OS than those with upregulation of *BICD1* (*n* = 164, 32.3%) (median survival: 6.263 years, 5-year survival: 61.8%, adjusted HR = 0.932) (*p* = 0.011254). In LGG patients with wild-type *IDH1* (*n* = 114, 22.4%), although the difference was not statically significant in the OS between the two subgroups stratified by *BICD1* expression (*p* = 0.122834), patients with downregulation of *BICD1* (*n* = 24, 4.7%) (5-year survival: 59.9%, adjusted HR = 1.868) still had a better OS than those with upregulation of *BICD1* (*n* = 90, 17.7%) (median survival: 1.992 years, 5-year survival: 24.9%, adjusted HR = 3.205) (Figure 5D). More importantly, LGG patients with *IDH1* mutations and downregulation of *BICD1* had a better OS (median survival: 8.186 years, 5-year survival: 80.3%, Figure 5D) than those with only *IDH1* mutations (median survival: 7.964 years, 5-year survival: 71.1%, Figure 5B). Our results highlighted that incorporating *BICD1* expression with *IDH1* mutation status improved the prognostic accuracy of *IDH1* status when it served as a prognostic marker of LGGs.

LGG patients were stratified into four subgroups according to their *EGFR* status and *BICD1* expression. In patients with wild-type *EGFR* (*n* = 473, 93.1%), those with downregulation of *BICD1* (*n* = 249, 49.0%) (median survival: 8.186 years, 5-year survival: 79.8%, adjusted HR = 0.480) had a significantly better OS than those with upregulation of *BICD1* (*n* = 224, 44.1%) (median survival: 6.252 years, 5-year survival: 55.7%, adjusted HR = 1.213) (*p* = 0.000038) (Figure 5E). Incorporating *BICD1* expression successfully stratified LGG patients with wild-type *EGFR* into two subgroups with a statistically significant difference in their OS, and LGG patients with wild-type *EGFR* and downregulation of *BICD1* had a better OS (median survival: 8.186 years, 5-year survival: 79.8%, Figure 5E) than those only with wild-type *EGFR* (median survival: 7.877 years, 5-year survival: 67.0%, Figure 5C). 

Our findings confirmed that *BICD1* expression is a useful marker for incorporation with *IDH1* or *EGFR* mutation status to stratify LGG patients into more clinically distinct subgroups and to obtain more accurate prognostic information.

### 3.5. BICD1 Expression Is an Independent Prognostic Factor in Patients with LGGs

In univariate Cox’s regression analyses, *EGFR* status (HR = 5.062, *p* = 5.215 × 10^−12^), *IDH1* status (HR = 4.445, *p* = 1.205 × 10^−15^), the WHO grade (HR = 3.314, *p* = 2.960 × 10^−9^), and patient age (HR = 2.889, *p* = 7.057 × 10^−8^) showed a stronger impact than *BICD1* expression (HR = 2.731, *p* = 0.000002) on the OS of LGG patients (Appendix A). In multivariate Cox’s regression analyses, *BICD1* expression was confirmed as an independent prognostic marker for indicating the OS of LGG patients (HR = 1.896, *p* = 0.004547) (Appendix A).

In univariate Cox’s regression analyses, the prognostic significance of *BICD1* expression was not as strong as that of *EGFR* status, *IDH1* status, the WHO grade, and patient age (Appendix A). In multivariate Cox’s regression analyses, conversely, *BICD1* expression (HR = 1.896, *p* = 0.004547) showed a significant and stronger impact than *IDH1* (HR = 1.687, *p* = 0.086014) and *EGFR* status (HR = 1.054, *p* = 0.857723) (Appendix A). It is important to notice that in univariate Cox’s regression analyses, 1p19q codeleted status did not show strong prognostic significance (HR = 2.602, *p* = 0.000067) (Appendix A). However, in multivariate Cox’s regression analyses, 1p19q codeleted status presented the highest prognostic significance among all variables (HR = 3.787, *p* = 0.000041) (Appendix A).

Our results confirmed the prognostic value of *BICD1* expression as an independent factor for indicating the outcome of LGG patients.

### 3.6. BICD1 Downregulation Probably Suppresses Signaling Pathways Related to Immune Checkpoints and Cancer Progression in LGGs

To explore the possible mechanism by which downregulation of *BICD1* contributed to a favorable outcome in LGG patients, we used the TCGA LGG cohort to analyze the correlation of *BICD1* expression with a couple of representative markers involved in several important pathways associated with cancer progression, including the immune checkpoint (PD-1 and PD-L1), *MET*, *STAT*, and *MTOR* pathways. The heatmap showed apparently positive correlations between downregulation of *BICD1* and a couple of markers, including *CD274* (PD-L1), *PDCD1* (PD-1), *GSK3B*, *HGF*, *MET*, *JAK2*, *STAT3*, *SOCS3*, *MTOR*, *RPS6KB1* (S6K), *EGFR*, *IDH1*, and *HIF1A*. Downregulation of *BICD1* also showed strong correlations with wild-type *EGFR* and *IDH1* mutations. Conversely, downregulation of *BICD1* revealed negative correlations with *DDIT4*, *ALDOA*, and *ALDH2* (Figure 6A).

The expression levels of *BICD1* showed the highest correlation with *CD274*(PD-L1) (Pearson’s r = 0.5415, *p* = 5.028 × 10^−40^) (Figure 6A,B). They were also significantly and positively correlated with *GSK3B* (Pearson’s r = 0.5106, *p* = 4.583 × 10^−35^) (Figure 6A,C), *HGF* (Pearson’s r = 0.4940, *p* = 1.322 × 10^−32^), *MET* (Pearson’s r = 0.3317, *p* = 1.659 × 10^−14^) (Figure 6A,D), *JAK2* (Pearson’s r = 0.4142, *p* = 1.808 × 10^−22^), *STAT3* (Pearson’s r = 0.4898, *P* = 5.332 × 10^−32^), *SOCS3* (Pearson’s r = 0.3624, *p* = 3.303 × 10^−17^) (Figure 6A,E), *MTOR* (Pearson’s r = 0.4003, *p* = 5.650 × 10^−21^) (Figure 6A,F), *EGFR* (Pearson’s r = 0.3743, *p* = 2.465 × 10^−18^) (Figure 6A,G), *IDH1* (Pearson’s r = 0.2487, *p* = 1.334 × 10^−8^) (Figure 6A,H), and *HIF1A* (Pearson’s r = 0.2270, *p* = 2.322 × 10^−7^) (Figure 6A,I). Conversely, the expression levels of *BICD1* were significantly and negatively correlated with *ALDH2* (Pearson’s r = −0.5531, *p* = 4.875 × 10^−42^) (Figure 6A,J).

Our analyses identified the high correlations between the downregulation of *BICD1* and the decreased levels of a couple of markers representing several important pathways associated with cancer progression, including immune checkpoint (PD-L1 related), *HGF*-*MET*, *JAK2-STAT3-SOCS3*, *GSK3B*, *MTOR-S6K*, *HIF1A*, *IDH1*, and *EGFR* pathways. Our findings may possibly explain why downregulation of *BICD1* contributed to a favorable outcome in LGG patients by connecting it with the suppression of pathways associated with cancer progression.

### 3.7. The Prognostic Significance of BICD1 Downregulation Is Validated by Another LGG Cohort from the CCGA Database

To ensure the consistency of our previous findings in the TCGA LGG cohort with other LGG datasets, the LGG (Grades II + III) patients in the CGGA dataset (*n* = 420) were used as a validation set for further correlation and survival analyses. The *t*-test analyses revealed that the expression levels of *BICD1* were significantly lower in LGGs with a lower WHO grade (grade II), *IDH1* mutations, 1p19q codeletions, and alive status of patients in the CGGA dataset. Notably, the expression levels of *BICD1* were highest in LGGs with wild-type *IDH1* and not codeleted 1p19q (Figure 7A). These findings are consistent with our previous analytical results for the TCGA LGG cohort. However, *BICD1* expression levels showed a significant (*p* = 0.00692) and negative correlation (Pearson r = −0.08319) with patient age in LGG patients in the CGGA dataset (Appendix A), which was not consistent with our result in the TCGA LGG cohort (Figure 4D).

LGG patients in the CGGA dataset (*n* = 420) were stratified into two groups depending on *BICD1* expression (50% high, *n* = 210 vs. 50% low, *n* = 210) for Kaplan–Meier survival analyses. Consistent with our previous results, those with downregulation of *BICD1* had a significantly better OS than those with upregulation of *BICD1* (*p* = 5.993 × 10^−6^) (Figure 7B), and LGG patients with *IDH1* mutations (*n* = 288, 68.6%) had a significantly better OS (median survival: 7.386 years, 5-year survival: 60.7%) than those with wild-type *IDH1* (*n* = 94, 22.4%) (median survival: 2.134 years, 5-year survival: 37.6%) (*p* = 1.046 × 10^−6^) (Figure 7C). Our results confirmed downregulation of *BICD1* as a favorable prognostic marker of LGGs.

*BICD1* expression was incorporated with *IDH1* status to stratify LGG patients in the CGGA dataset (*n* = 420) into further distinct subgroups for survival analysis. In LGG patients with *IDH1* mutations (*n* = 288, 68.6%), those with downregulation of *BICD1* (*n* = 135, 32.1%) (5-year survival: 70.6%, adjusted HR=0.607) had a significantly better OS than those with upregulation of *BICD1* (*n* = 153, 36.4%) (5-year survival: 51.2%, adjusted HR = 1.141) (*p* = 0.000253), and in LGG patients with wild-type *IDH1* (*n* = 94, 22.4%), those with downregulation of *BICD1* (*n* = 41, 9.8%) (median survival: 2.841, 5-year survival: 45.5%, adjusted HR = 1.290) had a significantly better OS than those with upregulation of *BICD1* (*n* = 53, 12.6%) (median survival: 1.836, 5-year survival: 31.3%, adjusted HR = 2.410) (*p* = 1.0004 × 10^−10^) (Figure 7D). Consistent with our previous results, LGG patients with wild-type *IDH1* and upregulation of *BICD1* had the worst OS (median survival: 1.836, 5-year survival: 31.3%, adjusted HR = 2.410) and those with *IDH1* mutations and downregulation of *BICD1* had the best OS (5-year survival: 70.6%, adjusted HR = 0.607). More importantly, LGG patients with *IDH1* mutations and downregulation of *BICD1* had a better OS (*n* = 135, 5-year survival: 70.6%, Figure 7D) than those only with *IDH1* mutations (*n* = 288, 5-year survival: 60.7%, Figure 7C). Our results confirmed that incorporating *BICD1* expression improves the prognostic accuracy of *IDH1* mutations when serving as a prognostic marker of LGGs.

The expression level of *BICD1* also showed a positive and significant correlation with that of *CD274* (PD-L1) (Pearson’s r = 0.2665, *p* < 0.0001) in LGGs in the CGGA dataset (Figure 7E). The gene expression profiles of LGG cell lines (*n* = 8) in the CCLE dataset were used as another validation set for confirming the correlation between downregulation of *BICD1* and that of *CD274* (PD-L1). The expression level of *BICD1* in LGG cell lines showed a positive and significant correlation with that of *CD274* (PD-L1) (Pearson’s r = 0.7549, *p* = 0.0304) in the CCLE dataset (Figure 7F).

Our findings confirmed that downregulation of *BICD1* is a favorable prognostic marker and is highly correlated with downregulation of several important markers associated with cancer progression, including *CD274* (PD-L1), in LGGs in the CGGA and CCLE datasets.

## 4. Discussion

The BICD Cargo Adaptor 1 (BICD1), encoded by *BICD1* gene and previously named *Bicaudal D* (*Drosophila*) *homolog 1*, was initially identified in Drosophila [16]. The *BICD1* and *BICD2* genes are two human homologues of the *Drosophila Bicaudal-D* gene. The *Drosophila Bicaudal-D* gene plays a role in drosophila oocyte differentiation [17]. In drosophila, mutations in the *Drosophila Bicaudal-D* gene result in a double abdomen, or a bicaudal (means “two-tailed”) phenotype of the drosophila embryo [18]. Subsequently, this gene was named “*Bicaudal-D*” (*BICD*) due to the striking phenotype when it was mutated [19]. In humans, naturally occurring variants in this gene are associated with short telomere length [20] and emphysema [21].

The *BICD1* gene encodes an adaptor protein that belongs to the *Bicaudal-D* family of dynein cargo adaptors. This protein acts as an intracellular cargo transport cofactor that regulates the microtubule-based transport of cargo onto the dynein motor complex [22]. It is also involved in cargo transport between the Golgi apparatus and endoplasmic reticulum and the recruitment of the dynein –dynactin complex [23]. Intracellular cargo transport via vesicle trafficking has been linked to tumor growth, tumor invasion, neo-angiogenesis, oncogenic transformation, and modulation of the immune response [24]. Recently, microtubule-associated factors were reported to be involved in hypoxia inducible factor 1A (HIF1A) nuclear translocation [25], and the BICD Cargo Adaptor 1 protein was shown to be a novel factor regulating the translocation of the HIF1A protein into nucleus [15].

The clinical significance of *BICD* expression in cancers was rarely reported before. A recent study demonstrated that upregulation of the *BICD1* gene is a predictor for poor prognosis and poor response to Temozolomide (TMZ) in glioblastoma (GBM) patients [26]. Another study also reported that BICD1 functions as a prognostic biomarker and promotes hepatocellular carcinoma (HCC) progression [27]. To date, there has been no report concerning the clinical relevance of *BICD1* downregulation in LGGs. Our study is the first to identify the novel characteristics of *BICD1* downregulation in LGGs and validate it as a potential biomarker for predicting a favorable prognosis of LGG patients.

In this study, we showed that downregulation of *BICD1* was significantly correlated with a couple of benign clinicopathological features in LGG patients, including a lower WHO grade (grade II), mutations in *IDH1*, *TP53*, and *ATRX*, wild-type *EGFR*, a younger patient age (≤40 years), and a higher KPS. In addition, the Kaplan–Meier survival analysis revealed that downregulation of *BICD1* predicts a better overall survival of LGG patients. All of our results were validated in two validation sets (LGG patients in the CGGA dataset and LGG cell lines in the CCLE dataset), which confirmed that downregulation of *BICD1* could be a potential biomarker for predicting a favorable prognosis of LGG patients.

Although *IDH1* mutations and 1p19q codeletions are both well-known markers for indicating a favorable outcome of LGG patients [28,29], the mechanism by which *IDH1* mutations or 1p19q codeletions contribute to a favorable outcome of LGGs is still not clearly elucidated. *IDH1* mutations are known to play a role in affecting cancer metabolism [30] and have been validated as the most significant biomarker for predicting a longer overall survival time in LGG patients (Figure 5B and Figure 7C). In our results, downregulation of *BICD1* was shown to be significantly correlated with *IDH1* mutations (Figure 4B). Although *BICD1* expression was not significantly corelated with 1p19q codeleted status, it was significantly upregulated in *IDH1* wild-type LGGs with *EGFR* mutations, which was the subgroup of LGGs with the poorest prognosis, and could be considered as glioblastoma (grade IV glioma), rather than LGGs, according to the 2016 WHO classification of CNS tumors (Figure 4C). Based on the previous finding that BICD1 promotes the translocation of HIF1A into the nucleus [15], we could try to answer the question of why *IDH1* mutations and 1p19q codeletions contributed to a favorable outcome in LGG patients by connecting *IDH1* mutations or 1p19q codeletions with the HIF1A pathway via the intermediate regulator, BICD1. In addition to *IDH1* mutations, *BICD1* downregulation was shown to significantly correlate with wild-type *EGFR* and mutations in *TP53* and *ATRX* (Figure 4B). Although mutations in *TP53* and *ATRX* are known molecular markers of LGGs, they did not have significant impact on the overall survival of LGG patient (*TP53* status: HR = 1.422, *p =* 0. 052633; *ATRX* status: HR = 1.408, *p =* 0. 067904, respectively) (Figure 2B), which also suggested that the expression status of *BICD1* had better prognostic value than the mutated status of *TP53* and *ATRX.*

Currently, the 2016 WHO classification of CNS tumors suggests the combined utilization of *IDH1* status, 1p19q codeleted status, and histological subtypes to provide more objective diagnosis and classification of CNS tumors [8]. However, there is still variation in clinical outcomes of patients who have the same *IDH1* or 1p19q codeleted status. Therefore, we attempted to incorporate *BICD1* expression with *IDH1* status to classify LGG patients into further distinct subgroups. With the combined utilization of *IDH1* status and *BICD1* expression, LGG patients with *IDH1* mutations could be stratified into two distinct subgroups with significant difference in their overall survival, and those with downregulation of *BICD1* had a significantly better overall survival than those with upregulation of *BICD1* (Figure 5D and Figure 7D). More importantly, LGG patients with *IDH1* mutations and downregulation of *BICD1* had a significantly better overall survival (Figure 5D) than those only with *IDH1* mutations (Figure 5B), and those with *IDH1* mutations and upregulation of *BICD1* had a significantly worse overall survival (Figure 7D) than those only with *IDH1* mutations (Figure 7C). Our results confirmed the prognostic value of *BICD1* expression as it could facilitate *IDH1* status to classify LGG patients into further clinically distinct subgroups, which may provide more accurate prognosis prediction of LGG patients than using *IDH1* status alone.

Despite the fact that *IDH1* status is the most well-known and powerful marker of LGGs, in our results, *EGFR* status was shown to have the highest impact on the overall survival of LGG patients (Figure 2B). In univariate Cox’s regression analyses, *EGFR* status (HR = 5.06), *IDH1* status (HR = 4.445), the WHO grade (HR = 3.314), patient age (HR = 2.889), and *BICD1* expression (HR = 2.731) were shown to have highly significant impact on the overall survival of LGG patients (Appendix A). In multivariate Cox’s regression analyses, patient age (HR = 2.673, *p* = 0.000011), the WHO grade (HR = 2.201, *p* = 0.000335), and *BICD1* expression (HR = 1.896, *p* = 0.004547) still had a significant impact on the overall survival of patients, which confirmed *BICD1* downregulation as an independent factor for a longer survival time in LGG patients (Appendix A). However, in multivariate Cox’s regression analyses, *IDH1* (HR = 1.687, *p* = 0.086014) and *EGFR* status (HR = 1.054, *p* = 0.857723) did not have a significant impact on the overall survival of LGG patients even though they had a highly significant impact on the overall survival of LGG patients in univariate Cox’s regression analyses. More interestingly, even though 1p19q codeleted status did not have a strong impact on the overall survival of LGG patients in univariate Cox’s regression analyses (HR = 2.602, *p* = 0.000067) (Appendix A), it presented the highest impact (HR = 3.787, *p* = 0.000041) on the overall survival of LGG patients among all variables in multivariate Cox’s regression analyses (Appendix A).

The underlying mechanism by which downregulation of *BICD1* resulted in a favorable outcome in LGG patients remains unclear. Based on the previously published study, upregulation of *BICD1*, which promotes the nuclear translocation of HIF1A and eventually enhances the transcriptional activity of HIF1A [15], could be a possible mechanism. To explore other possible mechanisms, we attempted to analyze the correlation of *BICD1* downregulation with a couple of markers involved in various pathways associated with cancer progression, including the immune checkpoint, *MET*, *STAT*, and *MTOR* pathways, in the TCGA LGG cohort. In our results, we observed positive and highly significant correlations of *BICD1* downregulation with decreased levels of *CD274* (PD-L1), *GSK3B*, *HGF*, and *STAT3* (Figure 6B,C,D), which may partially explain why LGG patients with downregulation of *BICD1* had a favorable outcome. Our findings were also validated in the CGGA (Figure 7E) and CCLE datasets (Figure 7F). It is important to identify the highly significant correlation between the expression level of *BICD1* and that of *CD274* (PD-L1). The *CD274* gene encodes the protein, Programmed Cell Death 1 Ligand 1 (PD-L1), which inactivates cytotoxic T-cells and helps tumor cells escape from immune damage by cytotoxic T-cells. Overexpression of *CD274* (PD-L1) has been identified as a poor prognostic marker in many types of human cancers, including colon cancer [31] and renal cell carcinoma [32,33]. Although its prognostic significance (HR = 1.767, *p* = 0.002327) was not as high as that of *BICD1* (HR = 2.731, *p* = 0.000002) in LGGs (Figure 2B), the strong correlation between *BICD1* downregulation and a decreased level of *CD274* (PD-L1) still highlighted the possibility that *BICD1* downregulation contributed to a favorable outcome in LGGs, which might be associated with an enhanced immune surveillance in cancer patients. Another interesting finding is that *BICD1* expression was shown to highly but negatively correlate with *ALDH2* expression (Pearson’s r = −0.5531, *p* = 4.875 × 10^−42^) (Figure 6A,J). *ALDH2* is an important tumor suppression gene in the pathogenesis of hepatocellular carcinoma [34], which may give us a hint that the favorable outcome of LGG patients with *BICD1* downregulation is probably due to upregulation of the tumor suppression gene, *ALDH2*.

A systemic review reported by Arash Ghaffari-Rafi and George Samandouras in 2020 showed that patients with oligodendroglioma have a better overall and progression-free survival on the basis of molecular subtypes of WHO grade II diffuse gliomas [35]. Accordingly, in multivariate Cox’s regression analyses, we found that patients with oligodendrogliomas had a significantly better overall survival than those with astrocytomas (*p* = 0.002872) (Figure 2B). Nevertheless, the statistical significance was not stronger than the impact of *BICD1* expression (*p* = 0.000002). Although *BICD1* expression in oligodendrogliomas was detected to be slightly higher than that of astrocytomas (Figure 3A), the difference was not statistically significant (*p* = 0.6061). Similarly, the correlation between the proportion of lower and higher BICD1 expression detected in the different histological subtypes of LGGs was not significant (Appendix A). Based on the results from Figure 2B, Figure 3A, Figure 6A, and Appendix A, we suggest that the impact of *BICD1* downregulation on a better overall survival in LGG patients is probably associated with a decreased level of PD-L1, *GSK3B*, *HGF*, or *STAT3* and is independent of LGG histological subtypes.

## 5. Conclusions

Our analyses suggested that *BICD1* downregulation could be a potential biomarker for indicating a favorable prognosis of LGG patients. Moreover, *BICD1* downregulation might be associated with an enhanced immune surveillance, in contrast to PD-L1-mediated immune suppression, and the suppressions of several signaling pathways, such as *JAK-STAT3-SOCS3*, *GSK3β-β-catenin*, *MTOR-S6K*, *HIF1A*, *HGF-MET*, *IDH1*, and *EGFR* in LGGs (Figure 8). These findings might provide hints for understanding the mechanism regarding the pathogenesis and progression in LGGs.

## Figures and Tables

**Figure 1 biology-10-00701-f001:**
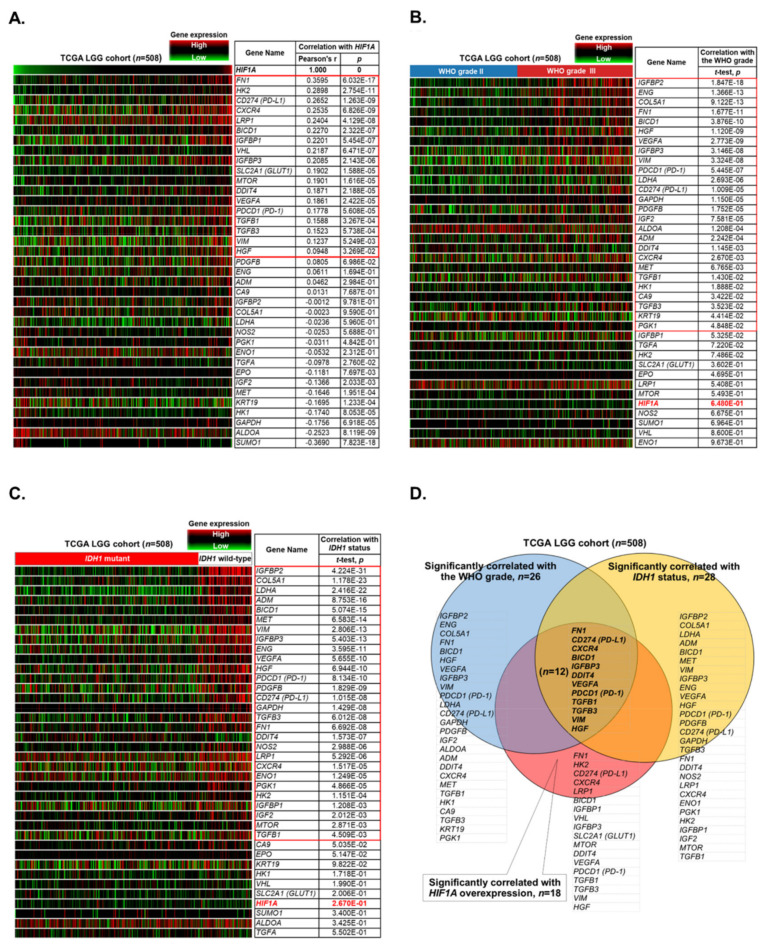
Identification of candidate biomarkers in the TCGA LGG database. (**A**). The heatmap revealed the correlations between the expression levels of 34 candidate genes and that of *HIF1A*. Those that were significantly and positively correlated with *HIF1A* upregulation were selected as candidate markers. (**B**). The heatmap revealed the correlations between the expression levels of 34 candidate genes and the WHO grade. Those that were significantly and differentially upregulated in grade III gliomas were selected as candidate markers. (**C**). The heatmap revealed the correlations between the expression levels of 34 candidate genes and *IDH1* status. Those that were significantly and differentially upregulated in LGGs with wild-type *IDH1* were selected as candidate markers. (**D**). The hierarchical method revealed the most qualified candidate markers that were compatible with all of our criteria.

**Figure 2 biology-10-00701-f002:**
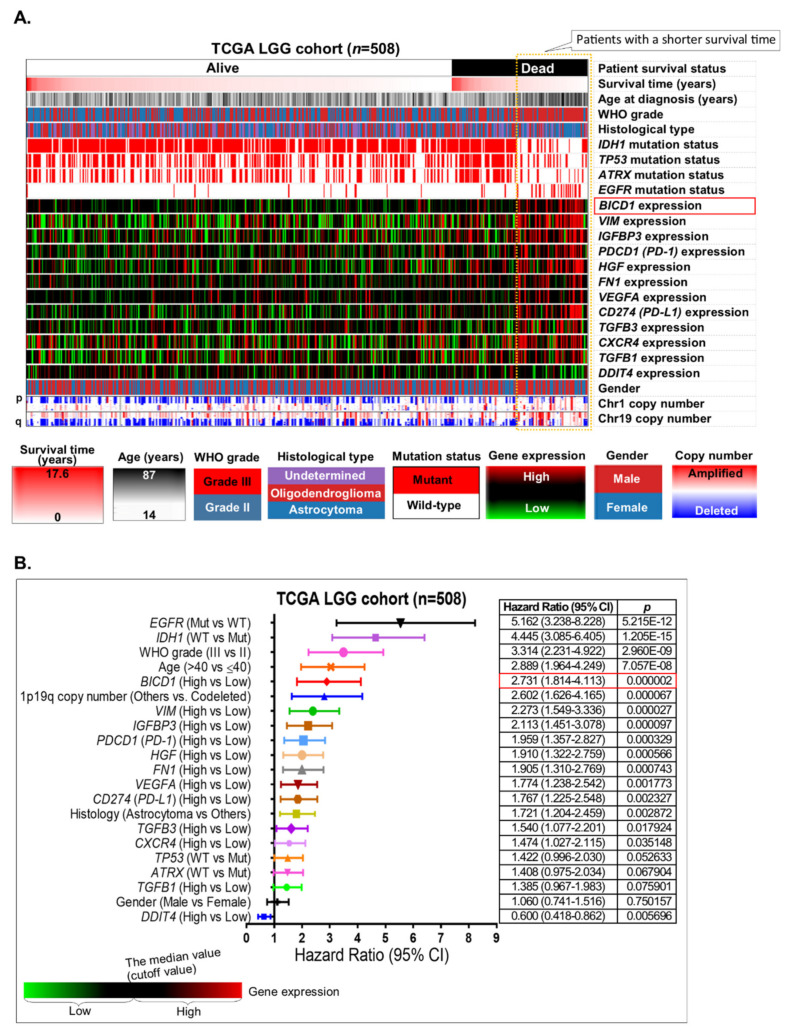
Identification of the putative marker that was upregulated in patients with a shorter overall survival and had highly prognostic significance in LGGs. (**A**). The heatmap revealed the correlations of patients’ overall survival with the clinicopathological features and the expression levels of the 12 candidate markers in the TCGA LGG cohort. (**B**). The prognostic significance and the hazard ratios of the 12 candidate markers were compared with those of the clinicopathological features to identify the putative marker. The median value of gene expression levels was used as a cutoff value to define high and low expressions of specific genes.

**Figure 3 biology-10-00701-f003:**
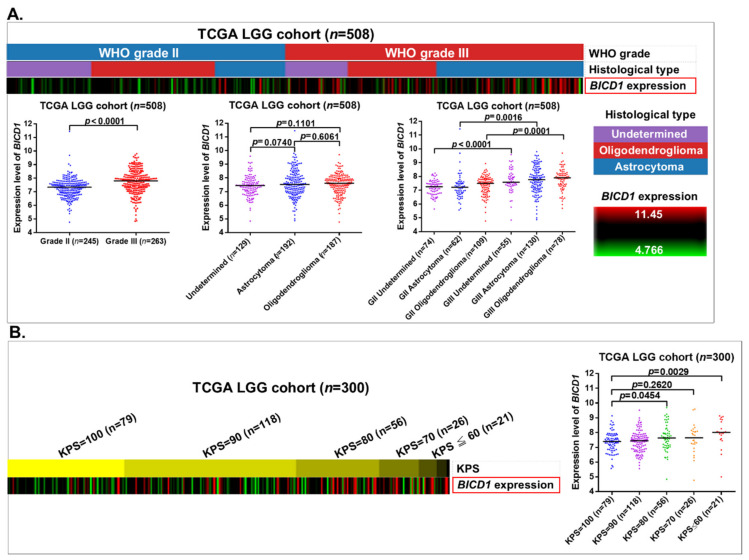
The correlations of *BICD1* downregulation with the WHO grade, histological subtypes, and the KPS of LGG patients. (**A**). The heatmap and the *t*-test analyses revealed the correlations of *BICD1* downregulation with the WHO grade and the histological subtypes of LGGs in the TCGA LGG cohort. (**B**). The heatmap and the *t*-test analysis revealed the correlations between the expression levels of *BICD1* and the KPS of patients in the TCGA LGG cohort.

**Figure 4 biology-10-00701-f004:**
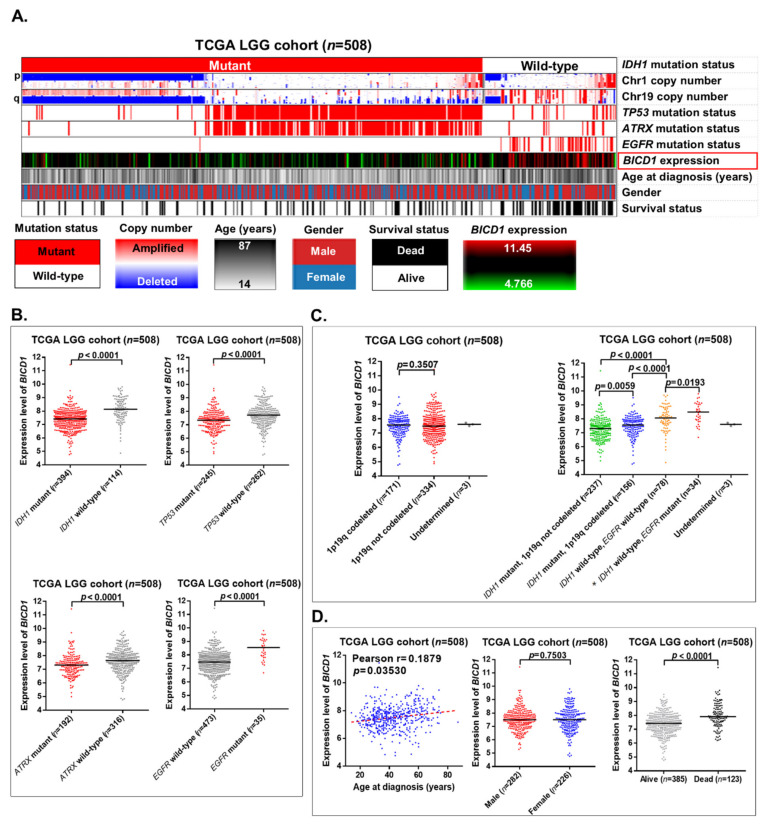
The correlations of BICD1 downregulation with chromosome 1p19q codeleted status, the mutation status of *IDH1*, *TP53*, *ATRX,* and *EGFR*, patient age, gender, and the survival status of LGG patients. (**A**). The heatmap revealed the correlations of BICD1 downregulation with 1p19q codeleted status, the mutation status of *IDH1*, *TP53*, *ATRX,* and *EGFR*, patient age, gender, and the survival status of patients in the TCGA LGG cohort. (**B**). The *t*-test analyses revealed the correlations between the expression levels of *BICD1* and the mutation status of *IDH1*, *TP53*, *ATRX,* and *EGFR*. (**C**). The *t*-test analyses revealed the correlation between the expression levels of *BICD1* and 1p19q codeleted status. The expression levels of *BICD1* were also demonstrated in four subgroups of LGGs stratified according to their *IDH1* status, 1p19q codeleted status, and *EGFR* status. * According to the 2016 WHO classification of CNS tumors, LGGs with wild-type *IDH1* and mutant *EGFR* could be considered to be glioblastoma (grade IV glioma) rather than LGGs (grade II and III gliomas). (**D**). The *t*-test and Pearson’s correlation analyses revealed the correlations of *BICD1* expression with patient age, gender, and their survival status.

**Figure 5 biology-10-00701-f005:**
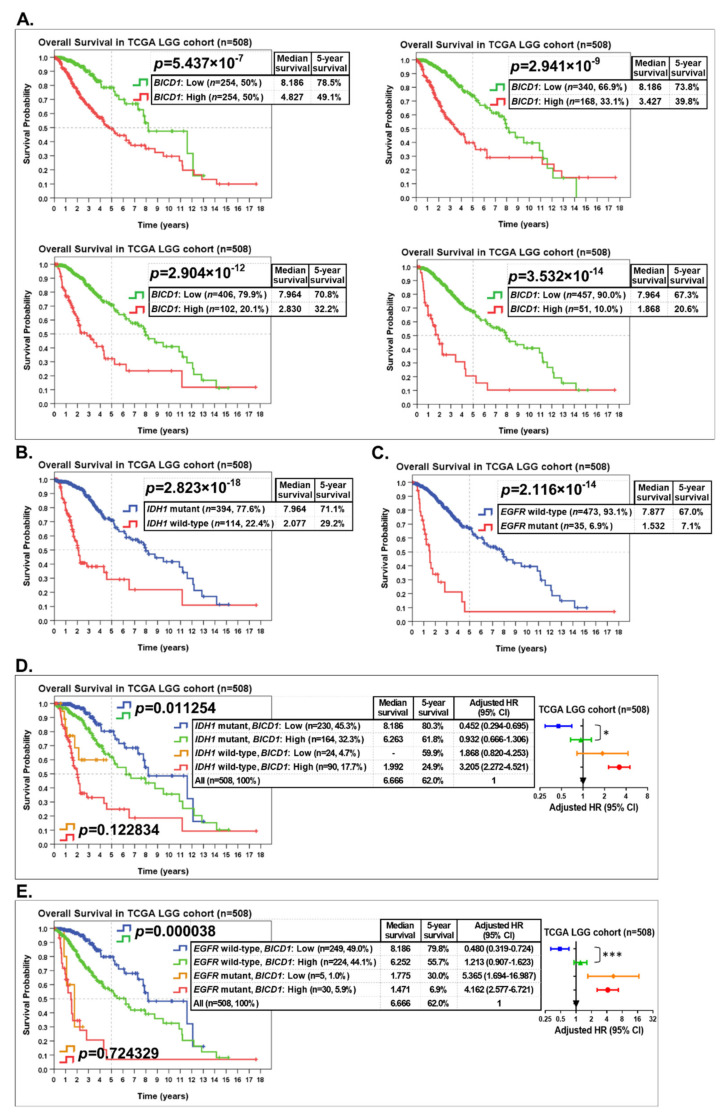
The prognostic role and significance of *BICD1* downregulation and the improvement in prognostic accuracy when *BICD1* expression was incorporated with *IDH1* or *EGFR* mutation status to stratify LGG patients. (**A**). The Kaplan–Meier survival analyses revealed the prognostic role and significance of *BICD1* downregulation when patients in the TCGA LGG cohort were stratified into two groups depending on their *BICD1* expression (50% high vs. 50% low, 33.1% high vs. 66.9% low, 20.1% high vs. 79.9% low, and 10.0% high vs. 90.0% low, respectively). (**B**). The prognostic role and significance of *IDH1* status in LGGs. (**C**). The prognostic role and significance of *EGFR* status in LGGs. (**D**). The improvement in prognostic accuracy when *BICD1* expression was incorporated with *IDH1* status to stratify LGG patients into four subgroups. The four subgroups of LGGs stratified by *IDH1* status and *BICD1* expression were ranked according to their HRs. (**E**). The improvement in prognostic accuracy when *BICD1* expression was incorporated with *EGFR* status to stratify LGG patients into four subgroups. The four subgroups of LGGs stratified by *EGFR* status and *BICD1* expression were ranked according to their HRs. In (**D**,**E**), the symbols “*” and “***” denote statistical *p* < 0.05 and 0.001, respectively.

**Figure 6 biology-10-00701-f006:**
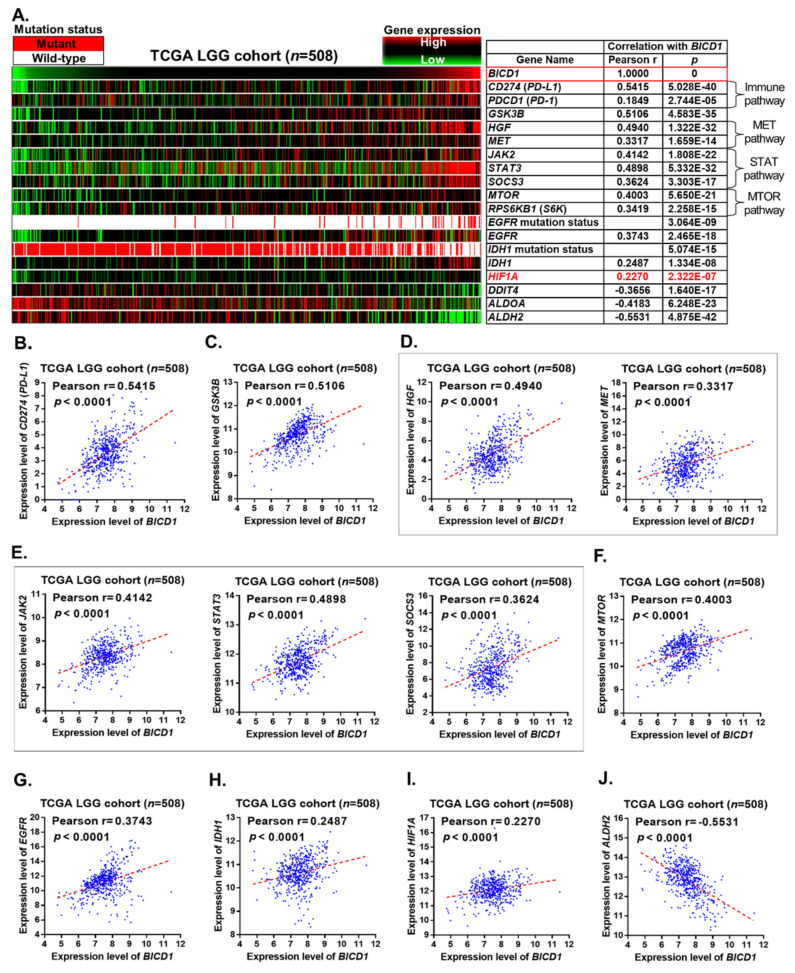
The correlations of *BICD1* downregulation with the well-known pathways associated with cancer progression. (**A**). The heatmap revealed the correlations of *BICD1* downregulation with the expression status of a couple of markers involved in various pathways associated with cancer progression in the TCGA LGG cohort. (**B**). Pearson’s correlation analysis revealed the correlation between the expression levels of *BICD1* and those of *CD274* (PD-L1). (**C**). The correlation between *BICD1* expression and *GSK3B* expression. (**D**). The correlations of *BICD1* expression with *HGF* expression and *MET* expression. (**E**). The correlations of *BICD1* expression with *JAK2* expression, *STAT3* expression, and *SOCS3* expression. (**F**). The correlation between *BICD1* expression and *MTOR* expression. (**G**). The correlation between *BICD1* expression and *EGFR* expression. (**H**). The correlation between *BICD1* expression and *IDH1* expression. (**I**). The correlation between *BICD1* expression and *HIF1A* expression. (**J**). The correlation between *BICD1* expression and *ALDH2* expression.

**Figure 7 biology-10-00701-f007:**
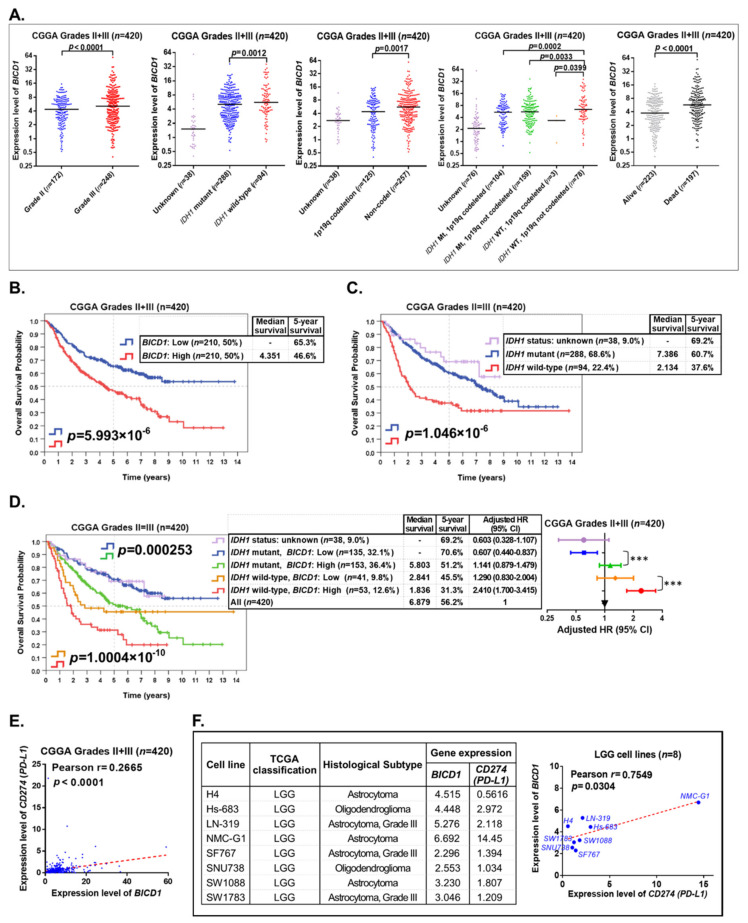
Validating the prognostic role and significance of *BICD1* downregulation and its correlations with the clinicopathological features in other LGG datasets. (**A**). The *t*-test analyses revealed the correlations of *BICD1* expression with the WHO grade, *IDH1* status, 1p19q codeleted status, and survival status in LGGs in the CGGA dataset. The differential expression of *BICD1* in LGG subgroups stratified by their *IDH1* status and 1p19q codeleted status is also shown. (**B**). The Kaplan–Meier survival analyses revealed the prognostic role and significance of *BICD1* downregulation in LGGs in the CGGA dataset. (**C**). The prognostic role and significance of *IDH1* status in LGGs in the CGGA dataset. (**D**). The improvement in prognostic accuracy when *BICD1* expression was incorporated with *IDH1* status to stratify LGG patients into four subgroups. The four LGG subgroups stratified by *IDH1* status and *BICD1* expression were ranked according to their HRs. The symbol “***” denotes statistical *p* < 0.001. (**E**). Pearson’s correlation analyses revealed the correlation between the expression level of *BICD1* and that of *CD274* (PD-L1) in LGGs in the CGGA dataset. (**F**). The expression levels of *BICD1* and *CD274* (PD-L1) in eight LGG cell lines in the CCLE dataset. Pearson’s correlation analysis revealed the correlation between the expression level of *BICD1* and that of *CD274* (PD-L1).

**Figure 8 biology-10-00701-f008:**
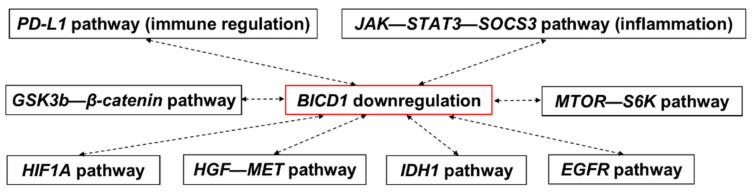
Possible pathways associated with *BICD1* downregulation in LGGs. A summary of the high correlations between *BICD1* downregulation and several well-known pathways associated with cancer progression.

## Data Availability

The clinicopathological data of patients in the TCGA LGG and CCGA cohort were downloaded from the TCGA Portal (http://www.xenabrowser.net/) and the CGGA website (http://www.cgga.org.cn/). The gene sequencing profiles of LGG cell lines in the CCLE dataset were also downloaded from the TCGA Portal (http://www.xenabrowser.net/). The cell lines we used in this manuscript were rechecked in the Cancer Model Passport website (https://cellmodelpassports.sanger.ac.uk/) to ensure all of them are LGG cell lines.

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
