# Peer review of "BICD Cargo Adaptor 1 (BICD1) Downregulation Correlates with a Decreased Level of PD-L1 and Predicts a Favorable Prognosis in Patients with IDH1-Mutant Lower-Grade Gliomas"

_biology, 2021, doi:10.3390/biology10080701_

Round 1

Reviewer 1 Report

The authors have addressed the comments satisfactorily. The paper can be accepted without any further changes required from the authors. 

Author Response

We sincerely thank Reviewer 1 for the positive comment.

Reviewer 2 Report

Dear Authors,

Your manuscript on the impact of BICD1 in LGGs prognostics is a comprehensive approach of a biomarker analysis. The performed analysis is well done and technically sound. The only point of concern I have is your interpretation of the data. You mention BICD1 to be important for lower grade glioma, but your analysis shows that BICD1 expression is mainly upregulated in IDH wild-type tumors which are most probably not lower grade gliomas. Therefore my conclusion would be that BICD1 expression is a biomarker for glioblastoma. You show that BICD1 expression is slightly higher in oligodendroglioma, that it is in astrocytoma, but the clinical significance is not clear to me as oligodendroglioma normally have a better overall and progression free survival. So in conclusion I would suggest to reconsider the focus of your work and change the headline accordingly.

Minor remark:

- The cell lines you mention in figure 7 F are not very commonly used. Therefore I would highly recommend to give some more details on them, to show that they are in the focus of your analysis. To my knowledge GI1 is a gliosarcoma cell line. A gliosarcoma is a mixture of RTK I and MES GBM and never IDH mutant. Therefore I have some doubt that this cell line is a LGG cell line.  

Best regards

Author Response

This manuscript is a resubmission of an earlier submission. The following is a list of the peer review reports and author responses from that submission.

Round 1

Reviewer 1 Report

In this manuscript, the authors propose a new prognostic biomarker BICD1, to better predict the clinical outcomes in patients with LGG. They analyzed data from two separate datasets to show that upregulation of BICD1 significantly correlated with several known-markers of poor prognosis in LGG.

Overall, the manuscript is comprehensively written and the data are thoroughly analyzed and presented. However, some minor issues need to be addressed/ clarified:

Line 101: Can you provide more information about this cohort here?- Like the molecular subtypes and the mutation status of this cohort.

Fig 5: Did you stratify LGG patients with ATRX and TP53 mutation based on BICD1 upregulation? According to Fi 4A BICD1 expression was higher in these subtypes as well.

Sections 2.5: Can you define BICD1 upregulation more clearly? In other words what cutoff are you using for BICD1 upregulation in cox regression analysis?

Typo in line 285: Change “Tn” to “To

Section 2.7: Can you check for correlation between BICD1 expression and patient age in another database ? (for eg., CCLE)

Reviewer 2 Report

Dear Authors,

Your manuscript on the impact of BICD1 in LGGs prognostics is based on an interesting approach to evaluate biomarkers. The performed analysis is well done, technically sound and presented in an appropriate way. Nevertheless, I do see one major flaw in the whole analysis which compromises your data. The databases you use are based on the old WHO classification (before 2016) and by this are known to include biological distinct groups. Especially the IDH wild-type glioma are known to be far more aggressive tumors and by this not to be graded WHO °II or III, but WHO °IV. This is also very obvious in your data. In addition the group of oligoastrocytoma is no longer be considered for the tumors that are inside this group in the databases. There are a very few exceptions to these two groups, but they are so rare that they can be neglected. What is shown in your manuscript is that there is a difference between IDH-mutant and IDH wild type tumors which is already known and well established. In addition the other biomarker correlations you show are also know and to my impression are based on the fundamentally different biology of the tumors analyzed.

To get meaningful data I highly recommend to stick to the established classification, which is easy to implement with the data available in the databases, and analyze BICD1 expression within these established groups. If you can show significant impact in the different groups I do believe that your contribution will have impact in the field.

There are some typos throughout the manuscript and a few points where language adjustment is needed. In supplementary figure 5, there are black boxes in the table and some values are missing.

Best regards

Reviewer 3 Report

As for definition Low grade glioma means glial tumors of grade 2. Anaplastic astrocytoma and other grade 3 tumors are high grade gliomas. Significant part of TCGA database are oligoastrocytomas. Today we know that these tumors are not so frequent, and probably are misdiagnosed, otherwise the diagnosis should be confirmed by molecular studies.